# BiMatting: Efficient Video Matting via Binarization

**Haotong Qin**[*1,2], **Lei Ke**[*2,3], **Xudong Ma**[1], **Martin Danelljan**[2], **Yu-Wing Tai**[4],
**Chi-Keung Tang**[3], **Xianglong Liu**[⊠1], **Fisher Yu**[2]
[1]Beihang University    [2]ETH Zürich    [3]HKUST    [4]Dartmouth College

## Abstract

Real-time video matting on edge devices faces significant computational resource constraints, limiting the widespread use of video matting in applications such as online conferences and short-form video production. Binarization is a powerful compression approach that greatly reduces computation and memory consumption by using 1-bit parameters and bitwise operations. However, binarization of the video matting model is not a straightforward process, and our empirical analysis has revealed two primary bottlenecks: severe representation degradation of the encoder and massive redundant computations of the decoder. To address these issues, we propose **BiMatting**, an accurate and efficient video matting model using binarization. Specifically, we construct shrinkable and dense topologies of the binarized encoder block to enhance the extracted representation. We sparsify the binarized units to reduce the low-information decoding computation. Through extensive experiments, we demonstrate that BiMatting outperforms other binarized video matting models, including state-of-the-art (SOTA) binarization methods, by a significant margin. Our approach even performs comparably to the full-precision counterpart in visual quality. Furthermore, BiMatting achieves remarkable savings of $12.4\times$ and $21.6\times$ in computation and storage, respectively, showcasing its potential and advantages in real-world resource-constrained scenarios. Our code and models are released at https://github.com/htqin/BiMatting.

## 1 Introduction

The success of deep neural networks has led to remarkable advancements in computer vision tasks, including video matting [1, 2, 3, 4, 5, 6, 1, 7, 8, 9, 10, 11, 12, 13, 14, 15, 16]. However, many practical applications based on deep networks require real-time processing with minimal latency, which is challenging due to the high computational and storage demands. To address this issue, researchers have developed lightweight video matting algorithms such as Robust Video Matting (RVM) [1] and BGMv2 [10]. While these methods achieve significant speedups and memory reductions, they still rely on expensive floating-point operations, leaving room for further compression.

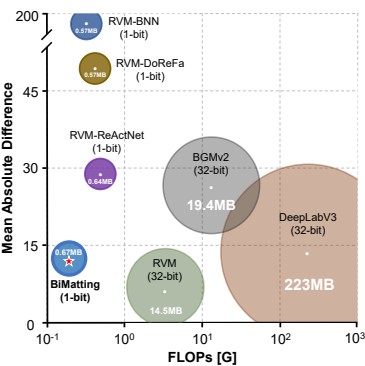

Figure 1: BiMatting enjoys impressive computation/storage savings while surpassing SOTA 1-bit and some 32-bit models in accuracy.

One of the most promising approaches to improving the efficiency of neural networks is network binarization [17, 18, 19, 20, 21, 22, 23, 24, 25, 26]. Binary neural networks (BNNs) have emerged as a highly effective technique for optimizing neural networks by reducing parameter bit width to 1-bit. BNNs leverage compact binary parameters that occupy less memory space and use efficient bitwise operations, which are much less computationally expensive than floating-point operations. By employing the binarization approach, researchers can significantly reduce the computational and storage demands of video matting applications.

---

* equal contribution    ⊠ corresponding author

37th Conference on Neural Information Processing Systems (NeurIPS 2023).

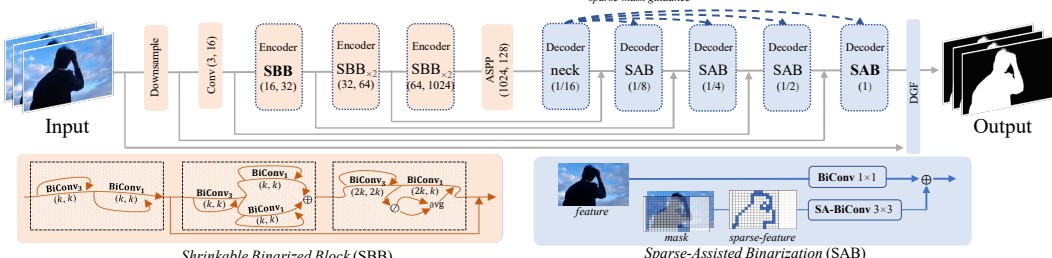

Figure 2: BiMatting overview. We apply Shrinkable Binarized Block (SBB) for the accurate encoder and Sparse-Assisted Binarization (SAB) for the efficient decoder. Arrows → indicate the flow of features, ⊕, ⊘, and avg indicate concatenation, splitting, and averaging of features, respectively.

Despite that the generic binarization methods have been studied extensively by the binarization community, the direct binarization for existing video matting models still leaves severe accuracy and efficiency bottlenecks. First, to construct an efficient architecture, many video matting methods contain encoders with lightweight full-precision backbones to obtain intermediate features. For example, MobileNet architectures are widely applied as lightweight extractors in several video matting models to provide strong features [1, 11]. However, this practice introduces an accuracy bottleneck in the binarized video matting model. Our analysis shows that the computing units and topology of existing lightweight encoder architectures are unfriendly to binarization, leading to severe degradation in the quality of intermediate features extracted by its binarized counterpart. Second, we identify that the architecture of the existing video matting decoder [1] causes immense redundant computation at the multi-scale decode stage. Intensive computation is still needed in large portions of certain foreground or background regions (spatial regions outside unknown parts of the trimap) of the high-resolution feature maps. This weakens the savings brought by binarization, which hinders efficiency improvements in the context of extremely compressed bit-width.

In this paper, we provide empirical studies of the above-mentioned bottlenecks. This leads us to propose **BiMatting**, a **Bi**narized model for accurate and efficient video **Matting** (overview in Fig. 2). To tackle the accuracy bottleneck, we first investigate the limitations of existing binarized encoders in representation extraction. Then, we propose *Shrinkable Binarized Block* (SBB), which follows a binarization-friendly computation-dense paradigm to construct a flexible block structure. SBB effectively extracts high-quality features with improved topology and operators. In addition, given a reliable binarized encoder, we further develop *Sparse-Assisted Binarization* (SAB) to effectively reduce the computational consumption of the binarized decoder. The overall burden is thus greatly reduced. SAB removes repeated computation by spatial sparseness [27] while preserving the accuracy of binarized units, resulting in a notable computation reduction without compromising performance.

Our BiMatting is the first binarization solution for video matting tasks, which surpasses 1-bit matting models using existing binarization algorithms by a significant margin [21, 17, 23, 18]. Notably, BiMatting even outperforms some full-precision models [28, 11] in terms of accuracy while being highly efficient. Our extensive experiments on fundamental tasks across VideoMatte240K (VM) [10], Distinctions-646 (D646) [29], and Adobe Image Matting (AIM) [30] datasets demonstrate that the advantages of BiMatting are task-independent. In addition, our SBB and SAB components are highly efficient, allowing BiMatting to achieve 12.4× FLOPs and 21.6× storage savings compared to the full-precision counterpart, leading a promising way for the video matting on edge scenarios (Fig. 1).

## 2   A Baseline for Binarized Video Matting

In this section, we first build a baseline to study the binarized video matting model. The baseline is based on straightforward binarization and an existing lightweight video matting architecture.

### 2.1   Binarization Framework

Binarization applies the sign function to compress weights and activations to 1-bit for computing units of the binarized network [17, 19, 20, 22, 31], and the propagation process can be expressed as

$$\text{sign}(\boldsymbol{x}) = \begin{cases} +1, & \text{if } \boldsymbol{x} \geq 0 \\ -1, & \text{otherwise} \end{cases}, \qquad \frac{\partial \mathcal{L}}{\partial \boldsymbol{x}} = \begin{cases} \frac{\partial \mathcal{L}}{\partial \text{sign}(\boldsymbol{x})}, & \text{if } x \in (-1, 1) \\ 0, & \text{otherwise} \end{cases}, \qquad (1)$$

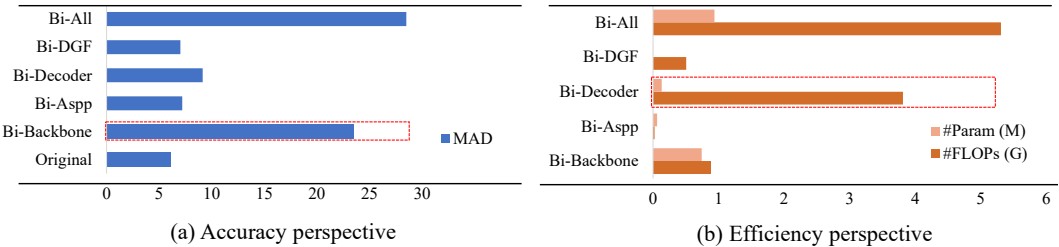

(a) Accuracy perspective          (b) Efficiency perspective

Figure 3: Analysis of bottlenecks for the binarized video matting baseline respectively in accuracy and efficiency on the VM dataset. We report (a) the MAD metrics of binarizing each component, and (b) the FLOPs and storage consumption of each component in the binarized baseline.

where $\mathcal{L}$ as the loss function. To build a strong binarized baseline by existing techniques, we apply floating-point scaling factors for weights $\boldsymbol{W}$ [19, 23] and learnable thresholds for activations $\boldsymbol{A}$ [23]:

$$\boldsymbol{B_w} = \text{sign}(\boldsymbol{W}), \qquad \boldsymbol{B_a} = \text{sign}(\boldsymbol{A} - \tau), \qquad \boldsymbol{o} = \alpha\boldsymbol{B_w} \otimes \boldsymbol{B_a}, \qquad (2)$$

where $\boldsymbol{B_w}$ and $\boldsymbol{B_a}$ denote the binarized weight and activation, respectively. The channel-wise $\alpha$ is obtained by $\text{mean}(|\boldsymbol{W}|)$, and the layerwise threshold $\tau$ is initialized as 0. Note that the Straight-Through Estimator (STE) [32] is uniformly applied to all experiments and binarization methods, approximating the gradient of the sign function as clip function (as the backward in Eq. (1)), since most other estimators consume unaffordable GPU memory [31, 22]. Binarization is applied to computation layers except for those who access original inputs or produce final outputs.

## 2.2 Video Matting Architecture

Matting model aims to break down a frame $I$ into a foreground $F$ and a background $B$, using an $\alpha$ coefficient to represent the linear combination of the two [1, 7],

$$I = \alpha F + (1 - \alpha)B. \qquad (3)$$

Compared to image matting [2, 3, 4, 5, 6, 33, 34, 35, 36], matting methods specifically developed for video sequence should be more effective in utilizing spatial-temporal information in videos [1, 7, 8, 9, 10, 11, 37, 38]. Among them, the recent Robust Video Matting (RVM) [1] method achieves the SOTA accuracy of the video matting task and stands out significantly in terms of efficiency. The architecture of RVM mainly comprises a lightweight MobileNetV3-based encoder (including backbone and Atrous Spatial Pyramid Pooling (ASPP) [39] module), a recurrent decoder, and a deep guided filter (DGF) [40] module. The concise architecture brings efficiency, even realizing real-time matting at high resolution. While RVM is one of the lightest video matting models available, there is still significant potential for compression in terms of reducing bit width, given the high cost of both the floating-point parameters and computations involved.

## 3 The Rise of BiMatting

### 3.1 Bottlenecks of Binarized Video Matting Baseline

Our goal is to achieve practical and resource-efficient video matting via binarization. However, the encoder and decoder of the binarized baseline pose accuracy and efficiency bottlenecks, respectively.

**From an accuracy perspective**, Fig. 3 (a) compares the accuracy drops on the VideoMatting (VM) dataset by binarizing every single part in the RVM model. Specifically, we find that binarizing the existing lightweight MobileNetV3 backbone in the encoder causes the most significant drop in accuracy among all parts, with a drop almost equivalent to directly binarizing the entire network (Bi-All 28.49 vs. Bi-Backbone 23.56 for MAD metric). In contrast, binarizing the ASPP, decoder, and DGF parts produce less harm to accuracy (less than a 3.05 MAD increase). Thus, improving the backbone of the encoder to make it more amenable to binarization is of the highest priority in addressing the accuracy drop in the binarized video matting model.

**From the efficiency perspective**, we show in Fig. 3 (b) the computation and storage usage for each part of the binarized video matting model to demonstrate the individual consumption in efficiency. Our analysis from an efficiency perspective reveals that the decoder consumes a significant amount

of computational resources, accounting for 71.8% (0.38G) FLOPs with only 12.1% parameters after binarization. Conversely, the binarized encoder has a higher parameter count (81.8%), but its FLOP consumption is only 16.6%. Negligible consumption is observed for the other parts. Based on these observations, it is evident that the computational redundancy of the decoder significantly impacts the acceleration performance of the overall model. Thus, there is ample room for computational optimization after direct binarization to further enhance the model's efficiency.

Given the observations from the aforementioned experiments, we find that the current baseline presents two major challenges: 1) the encoder's lightweight backbone is unsuitable for binarization and fails to generate practical features, and 2) the decoder continues to exhibit considerable computational redundancy even after binarization. Thus, we propose a binarization-friendly encoder and a computationally-efficient decoder to construct an accurate and efficient binarized matting model.

## 3.2 Shrinkable Binarized Block

### 3.2.1 Binarization-evoked Encoder Degradation

The binarized video matting baseline in Sec. 2 employs a lightweight MobileNetV3 as the encoder's backbone and directly binarizes its convolutional and linear layers. However, the extensive use of groupwise and pointwise convolutions presents significant challenges to binarization [41, 23], resulting in severe accuracy degradation. Fig. 3 shows that directly using binarized MobileNetV3 leads to over $4\times$ MAD metric degradation. On the other side, existing binarized networks are also far from the efficiency level of binarized MobileNetV3, making it hard to be transferred straightforwardly to construct the binarized matting model. For example, some binarized MobileNet-based models suffer at least 3-8$\times$ FLOPs than binarized MobileNetV3 [23, 41, 26]. Therefore, a new binarized backbone is necessary for feature extraction to achieve high-quality features for the decoder while keeping the model ultra-lightweight.

We first analyze the degradation of the binarized encoder in existing baselines. We note the binarized convolution as $\text{BiConv}_n(\cdot)$ or $\text{GBiConv}_n(\cdot)$, where $n$ is the kernel size of the convolution and G mark denotes groupwise. Their superscripts $^{\text{up}}$, $^{\text{eq}}$, and $^{\text{dn}}$ indicate the number of their output channels is greater than, equal to, or less than that of their input channels, respectively. Then the binarized MobileNetV3 block in the baseline's encoder can be expressed as

$$\begin{aligned}
\textit{MBV3 Block (1):} \quad &\boldsymbol{o} = \text{BiConv}_1^{\text{eq}}(\text{GBiConv}_n^{\text{eq}}(\text{BiConv}_1^{\text{eq}}(\boldsymbol{x}))) + \boldsymbol{x}, \quad s.t. \; c^{\boldsymbol{x}} = c^{\boldsymbol{o}} \\
\textit{MBV3 Block (2):} \quad &\boldsymbol{o} = \text{BiConv}_1^{\text{dn}}(\text{GBiConv}_n^{\text{eq}}(\text{BiConv}_1^{\text{up}}(\boldsymbol{x}))) + [c^{\boldsymbol{x}} = c^{\boldsymbol{o}}]\boldsymbol{x},
\end{aligned} \tag{4}$$

where $\boldsymbol{x}$ and $\boldsymbol{o}$ denote the input and output, respectively, kernel size $n \in \{3, 5\}$, and $[\cdot]$ denotes the *Iverson bracket* [42], evaluating to 1 if the condition inside the parentheses is true, and 0 otherwise. The batch normalization and the activation layers following convolutions are omitted.

Eq. (4) suggests at least two issues that impede accurate binarization. Firstly, all convolutions are groupwise or pointwise that have fewer parameters than regular convolutions, thus are sensitive to binarization and hard to recover from crashes caused by mutual influence. Secondly, utilizing only per-block short connection is insufficient since convolution-specific shortcuts are critical to achieving performance recovery [22, 23]. Several existing binarization methods aim to address the aforementioned issues, *e.g.*, applying regular convolutions instead of grouped ones [23, 26, 41], or creating shortcuts for channel-constant convolutions [23, 41]. Though these techniques may be effective, they are significantly more computational-expensive than directly binarized MobileNetV3 and still suffer representation loss. Therefore, it is necessary to develop stronger feature extraction encoders for binarized video matting models to address these limitations.

### 3.2.2 Shrinkable Binarized Block for Accurate Encoder

To mitigate the degradation of the encoder caused by binarization while keeping it lightweight, we present an efficient Shrinkable Binarized Block (SBB) for building a binarized video matting encoder, which is conducive to a flexible and lightweight architecture via channel-shrinkable design while retaining the representations and gradients.

Based on the analysis in Sec. 3.2.1, we determine that the crucial paradigm of an accurate binarized encoder is the computation-dense form of binarized block. In terms of topology, we ensure that every binarized convolution has a corresponding/dense connection to preserve the representation

accurately. In terms of the operator, introducing computation-dense regular convolutions can prevent groupwise and pointwise ones from becoming exclusive. Following this paradigm, SBB recovers the representation in the binarized block such that enables the backbone to effective feature extraction in flexible dimensional space.

Specifically, we first introduce the sub-blocks of SBB. Based on the above analysis, we find that the key element to constructing the flexible binarized architecture is to allow various sub-blocks in the feature extractor to freely adjust (increase, maintain, or shrink) the channel number of the outputs, while the shortcut should accompany with each binarized convolution to maintain representation. Therefore, the channel-shrunk SBB sub-block can be first constructed as follows:

$$\textit{Sub-SBB (1)} \quad \theta^{\mathrm{dn}}(\boldsymbol{x}): \quad \boldsymbol{o} = \mathrm{BiConv}_1^{\mathrm{dn}}(\boldsymbol{x}') + \mathrm{mean}\left(\boldsymbol{x}'^{\left(1, \frac{1}{2}c^{\boldsymbol{x}}\right)}, \boldsymbol{x}'^{\left(\frac{1}{2}c^{\boldsymbol{x}}, c^{\boldsymbol{x}}\right)}\right),$$
$$\boldsymbol{x}' = \mathrm{BiConv}_3^{\mathrm{eq}}(\boldsymbol{x}) + \boldsymbol{x}, \quad \textit{s.t. } c^{\boldsymbol{x}} = 2c^{\boldsymbol{o}}, \tag{5}$$

where $\boldsymbol{x}'^{(m,n)}$ denotes taking the $m$ to $n$ channels of $\boldsymbol{x}'$. In the sub-block, we introduce shortcuts for channel-shrunk binarized convolution with channel splitting and averaging operations, allowing the channel to shrink while constructing a shortcut with negligible overhead. This design enables the binarized encoder to leverage computing units with higher output channels (such as 32 or 64) for extracting low-channel representations (16 or 32, correspondingly), resulting in dependable features for the decoder. Inspired by [23, 41], the channel-maintain and increased sub-blocks are as follows:

$$\textit{Sub-SBB (2)} \quad \theta^{\mathrm{eq}}(\boldsymbol{x}): \quad \boldsymbol{o} = \mathrm{BiConv}_1^{\mathrm{eq}}(\boldsymbol{x}') + \boldsymbol{x}', \quad \boldsymbol{x}' = \mathrm{BiConv}_3^{\mathrm{eq}}(\boldsymbol{x}) + \boldsymbol{x}, \quad \textit{s.t. } c^{\boldsymbol{x}} = c^{\boldsymbol{o}}$$
$$\textit{Sub-SBB (3)} \quad \theta^{\mathrm{up}}(\boldsymbol{x}): \quad \boldsymbol{o} = ((\mathrm{BiConv}_{1_1}^{\mathrm{eq}}(\boldsymbol{x}') + \boldsymbol{x}') \oplus (\mathrm{BiConv}_{1_2}^{\mathrm{eq}}(\boldsymbol{x}') + \boldsymbol{x}')), \tag{6}$$
$$\boldsymbol{x}' = \mathrm{BiConv}_3^{\mathrm{eq}}(\boldsymbol{x}) + \boldsymbol{x}, \quad \textit{s.t. } c^{\boldsymbol{x}} = c^{\boldsymbol{o}}/2$$

where $\oplus$ denotes the concatenate operation. Benefiting from the above sub-block variants, SBB can implement a flexible feature extraction architecture.

Next, we create SBB by assembling these sub-blocks. Each SBB is made up of three sub-blocks: the head, middle, and tail. The head and tail sub-blocks produce identical output channels, while the middle sub-block extracts feature in higher dimensions with double output channels:

$$\textbf{SBB}: \quad \boldsymbol{o} = \theta^{\mathrm{dn}} \cdot \theta^{\mathrm{up}}(\boldsymbol{x}') + \boldsymbol{x}', \quad \boldsymbol{x}' = \theta^{\mathrm{eq}}(\boldsymbol{x})[c^{\boldsymbol{x}} = c^{\boldsymbol{o}}] + \theta^{\mathrm{up}}(\boldsymbol{x})\left[c^{\boldsymbol{x}} = \frac{1}{2}c^{\boldsymbol{o}}\right]. \tag{7}$$

where $\boldsymbol{x}'$ is the output of the head sub-block that varies $\theta^{\mathrm{eq}}$ or $\theta^{\mathrm{up}}$ depending on the channel constraint in *Iverson brackets*. The binarized sub-blocks in the middle and at the end of the sequence respectively increase and decrease the feature dimension for thorough extraction, while also incorporating shortcut across sub-blocks that allows the representations to remain intact. Using the block-level crossing shortcut also mitigates the influence of the splitting and averaging procedure on the representations in the trail channel-contracted sub-block.

### 3.3 Sparse-Assisted Binarization

#### 3.3.1 Computational Decoder Redundancy

In Sec. 3.1, we discover that while binarizing the decoder prevents accuracy performance from deteriorating, it results in significant computational costs (greater than 71.8%) despite having a relatively small parameter size (less than 12.1%). This inefficiency creates a bottleneck for the binarized video matting model. This is attributed to the decoder where high-resolution features are computationally demanding. To illustrate, consider the decoder's output block, which uses two $3 \times 3$ convolutions to perform computations on the original scale's features. Since grouped convolutions that are unfriendly to binarization are not utilized here, the accuracy reduction of decoder binarization is not as severe as it is on the encoder. But the computation of this single block in the decoder (the last one in 5 decoder blocks) is even equivalent to 103% of the entire encoder in a binarized baseline.

However, it is worth noting that not all spatial computations are equally crucial. In the case of video matting, the majority of frames have large, uninterrupted foreground and background areas, which can usually be identified at a lower resolution. Nevertheless, the current decoder architecture repeats these computations at various scales, leading to significant computational overhead, particularly on high-resolution feature maps. This issue renders constructing an efficient decoder challenging, even

with the most aggressive 1-bit binarization approach. As a result, a fundamental redesign of the computation unit is required to create an efficient decoder that minimizes redundant computation while maintaining accurate matting results. Such a revamp should consider the context of binarization and focus on streamlining the computation process while preserving the quality of the output.

### 3.3.2 Sparse-Assisted Binarization for Efficient Decoder

To address the computational redundancy problem in the decoder of the binarized video matting model, we propose the Sparse-Assisted Binarization (SAB) method in BiMatting.

Fortunately, benefiting from the analysis in Sec. 3.3.1, the solution to the computational redundancy of the decoder is intuitive, that is, to reduce the repeated intensive computation of continuous regions in the decoder, especially at the high-resolution features. First, inspired by the sparse segmentation and spatial region classification of the trimap, we hypothesize that most computation inside the matting decoder should be used in the unknown alpha matting regions, instead of an equal distribution on the certain foreground/background regions of the whole image. To approximate such unknown regions, we compute the incoherent regions $M_{inc}$ following [27] using the low-resolution output mask $M \in \{0,1\}^{\frac{N}{16} \times \frac{N}{16}}$ from the first decoder block, where $N$ denotes the original input scale.

Incoherent regions are primarily found along object instance boundaries or high-frequency areas (blue grids of the sparse feature in Fig. 2), and the points inside the incoherent regions are considered to be uncertain in alpha estimation. The remaining regions outside of the incoherent regions can be skipped/reduced in computation. Since the features of this process are the lowest resolution among the 4 sets of features input by the decoder, we obtain the mask $M_{inc}$ at a low cost.

Then, we employ the detected mask $M_{inc}$ to transform the convolutions in high-resolution blocks into sparse-assisted binarized convolutions for efficient computation. Our designed masked sparse convolution is different from [27, 43] on performing pointwise attention for the whole incoherent regions, resulting in a more binarization-friendly [44] and computation-efficient design. Specifically, we transform the 3×3 convolution in the up-sample block of the decoder into:

$$SAB: \quad \boldsymbol{o} = \text{SA-BiConv}_3(\boldsymbol{x}; \text{bilinear}^k(M_{inc})) + \text{BiConv}_1(\boldsymbol{x}), \tag{8}$$

where $\text{SA-BiConv}_3(\cdot; M)$ represents the sparse-assisted binarized 3×3 convolution under the guidance of sparse mask $M_{inc}$, in which the weight and sparse activation are binarized, and $\text{bilinear}^k$ denotes a bilinear interpolation up-sampling operation, and the low-resolution sparse mask $M_{inc}$ is up-sampled by $k$ times, $k \in \{2, 4, 8, 16\}$. For this binarized convolution $\text{SA-BiConv}_3$, the implementation of sparse computation follows [45, 27] to skip $M_{inc}(x, y) = 0$ masked regions during inference. However, this does not change that the essence of this convolution is still a non-grouped 3×3 convolution, which retains binarization-friendliness in practice. Moreover, as in Eq. (8), besides the sparse-assisted binarized 3×3 convolution, we also apply one 1×1 binarized convolution layer to process extraction in the whole feature and fuse the output obtained to that of SA-BiConv to guide the finer predictions of continuous regions.

Our SAB first obtains a sparse binary mask prediction based on low-resolution features, which is then used to assist sparse binarization in the higher-resolution features of the decoder. As validated in the experiment section (Tab. 1), SAB greatly reduces the computation of the binarized video matting model by reducing computational FLOPs by 30% while maintaining the accuracy of video matting.

### 3.4 Training Pipeline of BiMatting

We present the training pipeline for our BiMatting model, which involves additional training steps and iterations to ensure complete convergence of the binarized video matting model, as compared to the training of the full-precision RVM counterpart [1]. The training pipeline of our BiMatting is comprised of two phases, namely the pre-training phase and the matting training phase.

**Pre-training phase:** In the pre-training phase, we trained the binarized SBB backbone on the ImageNet classification dataset for 200 epochs to obtain a well-pre-trained backbone. This pre-training phase made it easier to converge during the matting training phase. Moreover, since the direct binarization of the full-precision pre-trained model, such as MobileNetV3, can lead to almost crashing, we apply the full pre-training phase for all compared binarized video matting models.

Table 1: Ablation result of BiMatting on VM [10] dataset. We ablate SBB and SAB in BiMatting at the encoder and decoder, respectively. Binarized RVM with MobileNetV3 is used as the baseline.

| Model | Method | #Bit | #FLOPs$_{(G)}$ | #Param$_{(MB)}$ | MAD | MSE | Grad | Conn | dtSSD |
|-------|--------|------|---------|---------|-----|-----|------|------|-------|
| RVM$_{mbv3}$ | - | 32 | 4.57 | 14.5 | 6.08 | 1.47 | 0.88 | 0.41 | 1.36 |
| RVM$_{mbv3}$ | - | 1 | 0.55 | 0.64 | 28.49 | 18.16 | 6.80 | 3.74 | 3.64 |
| RVM$_{SBB}$ | SBB | 1 | 0.57 | 0.67 | 14.81 | 7.63 | 3.16 | 1.70 | 2.70 |
| RVM$_{SAB}$ | SAB | 1 | 0.35 | 0.67 | 189.13 | 184.33 | 15.01 | 27.39 | 3.65 |
| BiMatting (Ours) | SBB+SAB | 1 | 0.37 | 0.67 | 12.82 | 6.65 | 2.97 | 1.44 | 2.69 |

**Matting training phase:** This phase is binarization-aware and built following [1], divided into 4 stages. **Stage 1** involves training on the low-resolution VM dataset for 20 epochs without DGF, with $T = 15$ frames for quick updates. The SBB backbone's learning rate is set as $1e^{-4}$ and the rest as $2e^{-4}$. Additionally, the input resolution $h, w$ is sampled independently from 256-512px for improving robustness. In **Stage 2**, the network is trained with $T = 50$, with halved learning rate and 2 more epochs to enable learning of long-term dependencies. In **Stage 3**, the DGF module is attached, and 1 epoch is trained on both low-resolution long and high-resolution short sequences from the VM dataset. The low-resolution pass is $T = 40$, $h$ and $w$ are the same with stage 1 without DGF, while the high-resolution pass employs DGF with downsample factor $s = 0.25$, $\hat{T} = 6$, and $\hat{h}, \hat{w} \sim (1024, 2048)$. The learning rate of DGF is $2e^{-4}$, and that of the rest is $1e^{-5}$. In **Stage 4**, the network is trained for 5 epochs on D646 and AIM, increasing the decoder's learning rate to $5e^{-5}$.

## 4 Experiments

We extensively evaluate the accuracy and efficiency of our proposed BiMatting. We first ablate our method and illustrate the contributions of SBB and SAB on the VM [10] dataset. Then we compare BiMatting with binarized video matting models that utilize existing binarization techniques on VM [10], D646 [29], and AIM [30], where our designs excel and even outperform some full-precision video matting models. Concerning efficiency, BiMatting impressively reduces computational FLOPs and model size by $11.2\times$ and $21.6\times$, respectively. On metrics, we evaluate $\alpha$ using mean absolute difference (MAD), mean squared error (MSE), spatial gradient (Grad), and connectivity (Conn) for quality, and dtSSD for temporal coherence. We also measure pixels where $\alpha > 0$ by MSE for $F$ [1]. All stages of our experiments use batch size 4 splits across 4 Nvidia A100 GPUs.

### 4.1 Ablation Study

Tab. 1 demonstrates that the binarized video matting baseline experiences a significant drop in performance across all accuracy metrics for VM data recall. Despite its $22.7\times$ parameter compression, the model only realizes an $8.3\times$ computational savings in the efficiency metric. Upon substituting the encoder with our SBB alone, the binary model's accuracy is considerably restored, affirming that the encoder is the principal performance bottleneck in the baseline. However, the decoder's implementation of the efficient SAB does not fully resolve the performance bottleneck, where the accuracy of the binarized model is perilously close to collapsing. By integrating both of our contributions, both accuracy and efficiency performance is substantially enhanced. Notably, in BiMatting, combining these two improvements can produce a remarkable enhancement in accuracy performance, underscoring the importance that the decoder should concentrate on less but crucial representations to heighten model performance by providing high-quality features.

### 4.2 Comparison Results

To create the comparison benchmark, we combine test samples from the VM, D646, and AIM datasets with 20 video backgrounds and 20 image backgrounds, following the settings in [1, 47]. Each test clip contains 100 frames where motion augmentation is applied to image samples. We compare our BiMatting model with other video matting models that have been binarized using existing methods. These binarization methods include classical BNN [41] and DoReFa [18], as well as state-of-the-art (SOTA) ReActNet [23] and ReCU [21], where the latter two are considered as best practices for generic binarization [44]. In addition, we have also included results from some full-precision video matting methods for comparisons, such as the RVM [1] with MobileNetV3 [48] backbone (oracle), DeepLabV3 [28], and background-based BGMv2 [10] with MobileNetV2 [49] backbone. To ensure

Table 2: Low-resolution comparison on VM, D646, and AIM datasets. **Bold** indicates the best performance among binarized video matting models and $^{\dagger}$ indicates the results is crashed.

| Dataset | Method | #Bit | #FLOPs(G) | #Param(MB) | Alpha | | | | | FG |
| | | | | | MAD | MSE | Grad | Conn | dtSSD | MSE |
|---|---|---|---|---|---|---|---|---|---|---|
| VM | DeepLabV3 | 32 | 136.06 | 223.66 | 14.47 | 9.67 | 8.55 | 1.69 | 5.18 | - |
| 512×288 | BGMv2 | 32 | 8.46 | 19.4 | 25.19 | 19.63 | 2.28 | 3.26 | 2.74 | - |
| | RVM (oracle) | 32 | 4.57 | 14.5 | 6.08 | 1.47 | 0.88 | 0.41 | 1.36 | - |
| | RVM-BNN$^{\dagger}$ | 1 | 0.50 | 0.57 | 189.13 | 184.33 | 15.01 | 27.39 | 3.65 | - |
| | RVM-DoReFa | 1 | 0.52 | 0.57 | 51.64 | 34.50 | 8.85 | 7.14 | 4.09 | - |
| | RVM-ReCU$^{\dagger}$ | 1 | 0.52 | 0.64 | 189.13 | 184.33 | 15.01 | 27.39 | 3.65 | - |
| | RVM-ReAct | 1 | 0.55 | 0.64 | 28.49 | 18.16 | 6.80 | 3.74 | 3.64 | - |
| | BiMatting (Ours) | 1 | **0.37** | 0.67 | **12.82** | **6.65** | **2.97** | **1.42** | **2.69** | - |
| D646 | DeepLabV3 | 32 | 241.89 | 223.66 | 24.50 | 20.1 | 20.30 | 6.41 | 4.51 | - |
| 512×512 | BGMv2 | 32 | 16.48 | 19.4 | 43.62 | 38.84 | 5.41 | 11.32 | 3.08 | 2.60 |
| | RVM (oracle) | 32 | 8.12 | 14.5 | 7.28 | 3.01 | 2.81 | 1.83 | 1.01 | 2.93 |
| | RVM-BNN$^{\dagger}$ | 1 | 0.88 | 0.57 | 281.20 | 276.85 | 25.26 | 73.59 | 1.08 | 6.95 |
| | RVM-DoReFa | 1 | 0.92 | 0.57 | 133.63 | 116.69 | 17.09 | 35.08 | 2.58 | 6.97 |
| | RVM-ReCU$^{\dagger}$ | 1 | 0.92 | 0.64 | 281.20 | 276.85 | 25.26 | 73.59 | 1.08 | 6.95 |
| | RVM-ReAct | 1 | 0.97 | 0.64 | 56.41 | 43.10 | 14.05 | 14.85 | 2.56 | 6.85 |
| | BiMatting (Ours) | 1 | **0.66** | 0.67 | **32.74** | **24.48** | **9.34** | **8.62** | **2.21** | **5.86** |
| AIM | DeepLabV3 | 32 | 241.89 | 223.66 | 29.64 | 23.78 | 20.17 | 7.71 | 4.32 | - |
| 512×512 | BGMv2 | 32 | 16.48 | 19.4 | 44.61 | 39.08 | 5.54 | 11.60 | 2.69 | 3.31 |
| | RVM (oracle) | 32 | 8.12 | 14.5 | 14.84 | 8.93 | 4.35 | 3.83 | 1.01 | 5.01 |
| | RVM-BNN$^{\dagger}$ | 1 | 0.88 | 0.57 | 327.02 | 321.15 | 23.80 | 85.55 | 0.75 | 7.84 |
| | RVM-DoReFa | 1 | 0.92 | 0.57 | 129.29 | 107.79 | 17.31 | 34.18 | 2.62 | 7.85 |
| | RVM-ReCU$^{\dagger}$ | 1 | 0.92 | 0.64 | 327.02 | 321.15 | 23.80 | 85.55 | 0.75 | 7.84 |
| | RVM-ReAct | 1 | 0.97 | 0.64 | 59.90 | 44.08 | 14.32 | 15.90 | 2.37 | 8.00 |
| | BiMatting (Ours) | 1 | **0.66** | 0.67 | **35.17** | **26.53** | **9.42** | **9.24** | **1.82** | **7.00** |

Table 3: High-resolution comparison on VM, D646, and AIM datasets. $^{*}$ indicates using the officially released model directly [46].

| Dataset | Method | #Bit | #FLOPs(G) | #Param(MB) | SAD | MSE | Grad | dtSSD |
|---|---|---|---|---|---|---|---|---|
| VM | RVM | 32 | 4.15 | 14.5 | 6.57 | 1.93 | 10.55 | 1.90 |
| 1920×1080 | BGMv2$^{*}$ | 32 | 9.86 | 19.4 | 49.83 | 44.71 | 74.71 | 4.09 |
| | RVM-ReAct | 1 | 0.53 | 0.64 | 31.60 | 20.29 | 34.28 | 4.08 |
| | BiMatting (Ours) | 1 | **0.38** | 0.67 | **18.16** | **11.15** | **21.90** | **3.25** |
| D646 | RVM | 32 | 8.37 | 14.5 | 8.67 | 4.28 | 30.06 | 1.64 |
| 2048×2048 | BGMv2$^{*}$ | 32 | 15.19 | 19.4 | 57.40 | 52.00 | 149.20 | 2.56 |
| | RVM-ReAct | 1 | 1.07 | 0.64 | 57.38 | 42.14 | 71.24 | 3.03 |
| | BiMatting (Ours) | 1 | **0.77** | 0.67 | **52.85** | **44.08** | **61.60** | 3.12 |
| AIM | RVM | 32 | 8.37 | 14.5 | 14.89 | 9.01 | 34.97 | 1.71 |
| 2048×2048 | BGMv2$^{*}$ | 32 | 15.19 | 19.4 | 45.76 | 38.75 | 124.06 | 2.02 |
| | RVM-ReAct | 1 | 1.07 | 0.64 | 57.38 | 42.14 | 71.24 | 3.03 |
| | BiMatting (Ours) | 1 | **0.77** | 0.67 | **48.27** | **38.37** | **61.72** | **2.80** |

a fair comparison, we apply the exact same training pipeline to all binarized video matting networks as we did to BiMatting, as explained in Sec. 3.4. As for the full-precision video matting model, we follow the results reported in previous studies [1], unless otherwise specified.

Tab. 2 presents a comparison of methods that use low-resolution input. The findings indicate that applying BNN and ReCU directly to full-precision RVM leads to completely collapsed results. This is surprising given that the latter is among the SOTA binary methods, highlighting that binarizing existing video matting architectures is not a straightforward task. In contrast, our BiMatting model performs significantly better than all existing binarization models across all datasets, which predicts alpha with higher accuracy and consistency, resulting in more coherent and accurate performance. Further details will be presented in Sec. 4.3 with comprehensive visualizations. Furthermore, BiMatting even outperforms some 32-bit full-precision models when using only 1-bit limit bit width. For instance, BiMatting surpasses BGMv2 on VM, D646, and AIM datasets, as well as DeepLabV3 on

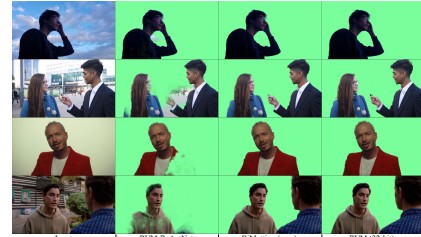 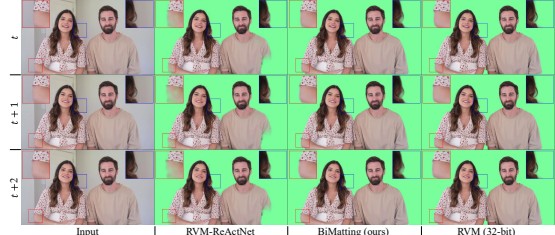

(a) Frame matting comparison.  (b) Temporal coherence comparison.

Figure 4: We compare our BiMatting with the existing SOTA binarized RVM-ReActNet and 32-bit full-precision (32-bit) RVM models to demonstrate its excellent accuracy.

VM datasets. These results demonstrate the enormous potential of binarization for efficient video matting. Tab. 3 presents a comparison between our BiMatting approach and other methods using high-resolution datasets. Our method consistently outperforms existing binarization models and BGMv2 across multiple metrics, demonstrating the robustness of BiMatting under varying resolutions.

Moreover, BiMatting has demonstrated its high potential for video matting in efficiency with impressive accuracy. As Tab. 2 and Tab. 3 show, in comparison to the full-precision counterpart (RVM), BiMatting achieves a computational FLOPs savings of 12.4 times and parameter savings of 21.6 times, making it the most promising solution for edge deployment. Moreover, BiMatting outperforms existing binarization methods, which offer significant acceleration advantages with only a tiny parameter overhead, such as a 0.1M increase compared to RVM-BNN.

### 4.3 Visualization

Fig. 4 and Fig. 5 show qualitative comparisons of natural videos. Fig. 4 compares our BiMatting model with RVM-ReActNet (the

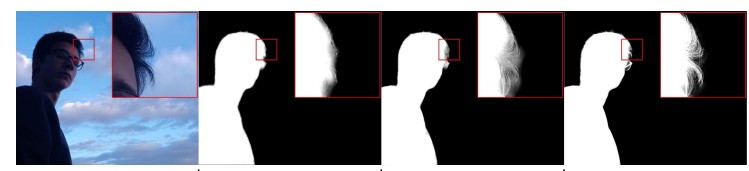

Input | RVM-ReActNet | BiMatting (ours) | RVM (32-bit)

Figure 5: Alpha detail comparison.

current SOTA binarized video matting model) and the full-precision RVM (can be seen as the 32-bit counterpart of BiMatting). In general, BiMatting's performance is close to the full-precision counterpart with greatly reduced resource consumption. Specifically, in Fig. 4(a), we conduct experiments on videos from diverse video frames. Our results reveal that BiMatting is more robust against semantic errors. Furthermore, BiMatting outperforms the other two models in matting edge regions. Fig. 4(b) compares temporal coherence, where our BiMatting consistently produces superior results, while RVM-ReActNet produces different areas of error. In Fig. 5, a comparison is made among the alpha predictions of various methods. Observe that our method outperforms the SOTA binarized video matting model by more accurately predicting intricate details, such as individual strands of hair.

**Limitation**. Though significantly improved, our BiMatting is not yet on par with its full-precision counterpart, the 32-bit RVM, particularly when it comes to local details such as the hair ends which tend to get blurred, while also favoring simpler backgrounds that lead to accurate matting results.

## 5 Conclusion

Our proposed model, BiMatting, is an efficient and accurate solution that utilizes binarization to achieve real-time video matting on edge devices constrained by computational resources. In this paper, we address the primary bottlenecks of binarization by constructing shrinkable and dense topologies for the binarized encoder block to enhance representation and sparsifying the binarized units to reduce redundant decoder computation. Our exhaustive experiments demonstrate that the proposed BiMatting outperforms existing binarized video matting models by a significant margin while producing a comparable performance to the full-precision counterpart in visual quality. BiMatting achieves significant savings in computation and storage, making it an attractive solution for real-world resource-constrained scenarios such as online conferences and short-form video production.

**Acknowledgement** This work was supported by the National Natural Science Foundation of China (No. 62022009), the State Key Laboratory of Software Development Environment (SKLSDE-2022ZX-23).

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

# BiMatting: Efficient Video Matting via Binarization

## 6 Overview

We offer further information in this supplementary material. Section 7 elaborates on our network architecture, providing detailed insights. In Section 8, we present additional results for detailed comparison and also showcase visual examples of our composited matting data samples. We highlight that video results are included in our supplementary material. We encourage readers to refer to our provided video for extensive matting results comparisons.

## 7 Details of Network Architecture

For the encoder (Table 4) constructed by our proposed SBB, it operates on individual frames to extract feature maps at various spatial scales ranging from $\frac{1}{2}$ to $\frac{1}{16}$. Within each SBB, the number of feature channels is either doubled or stays the same in the first sub-block, then doubled again in the second sub-block, and finally halved in the third sub-block. This design enables each SBB to carry out feature extraction in a channel dimension space that exceeds the input dimension. Meanwhile, the computation-dense structure guarantees the utilization of binarized convolutions to acquire high-quality features. While traditional full-precision MobileNetV3 backbones operate at $\frac{1}{32}$ scale, we made modifications to the last block. Specifically, we utilize convolutions with a dilation rate of 2 and a stride of 1, following the design principles from [1]. Furthermore, the final feature map ($\frac{1}{16}$ scale) is passed to the LR-ASPP module, which compresses it into 128 channels.

| Module | Conv$_{32bit}$ | SBB_1 | | | SBB_2 | | | SBB_3 | | |
|---|---|---|---|---|---|---|---|---|---|---|
| Sub-Module | | Sub-SBB (3) | Sub-SBB (3) | Sub-SBB (1) | Sub-SBB (3) | Sub-SBB (3) | Sub-SBB (1) | Sub-SBB (2) | Sub-SBB (3) | Sub-SBB (1) |
| In/Out Channel | (3, 16) | (16, 32) | (32, 64) | (64, 32) | (32, 64) | (64, 128) | (128, 64) | (64, 64) | (64, 128) | (128, 64) |
| Extracted Feature | $\frac{1}{2}$ | | | $\frac{1}{4}$ | | | | | | $\frac{1}{8}$ |
| Module | | SBB_4 | | | SBB_5 | | | | | ASPP |
| Sub-Module | | Sub-SBB (3) | Sub-SBB (3) | Sub-SBB (1) | Sub-SBB (2) | Sub-SBB (3) | Sub-SBB (1) | Sub-SBB (3) | Sub-SBB (3) | |
| In/Out Channel | | (64, 128) | (128, 256) | (256, 128) | (128, 128) | (128, 256) | (256, 128) | (128, 256) | (256, 1024) | (1024, 128) |
| Extracted Feature | | | | | | | | | $\frac{1}{16}$ | |

Table 4: The details in the encoder of our BiMatting, where "Feature Scale" indicates the scale of features extracted by this sub-block that is utilized by the decoder. Sub-SBB (1), (2), and (3) follow the notations of Eq. (5) and (6) in our paper to represent different types of sub-blocks.

For the decoder (Table 5), as mentioned in our paper, the SAB is employed in every decoder block except the first one to accelerate computations. The binary mask used by the SABs is obtained from the first non-sparse binarized block, which has the smallest feature scale and acquires the mask at a minimal cost. This design significantly improves the efficiency of the decoder.

| Module | BottleNeck | SAB_1 (Upsampling) | SAB_2 (Upsampling) | SAB_3 (Upsampling) | SAB_4 (Output) |
|---|---|---|---|---|---|
| Feature Scale | $\frac{1}{16}$ | $\frac{1}{8}$ | $\frac{1}{4}$ | $\frac{1}{2}$ | $\frac{1}{1}$ |
| Input Mask | | $M_{inc}$ | $M_{inc}$ | $M_{inc}$ | $M_{inc}$ |
| Produced Mask | $M_{inc}$ | | | | |

Table 5: The details in the decoder of our BiMatting, where "Extracted Feature" indicates that the features extracted by this sub-block are utilized by the decoder. Sub-SBB (1), (2), and (3) follow the notations of Eq. (5) and (6) in our paper to represent different types of sub-blocks. $M_{inc}$ is the sparse mask to guide the decoder computation mainly in "difficult" regions.

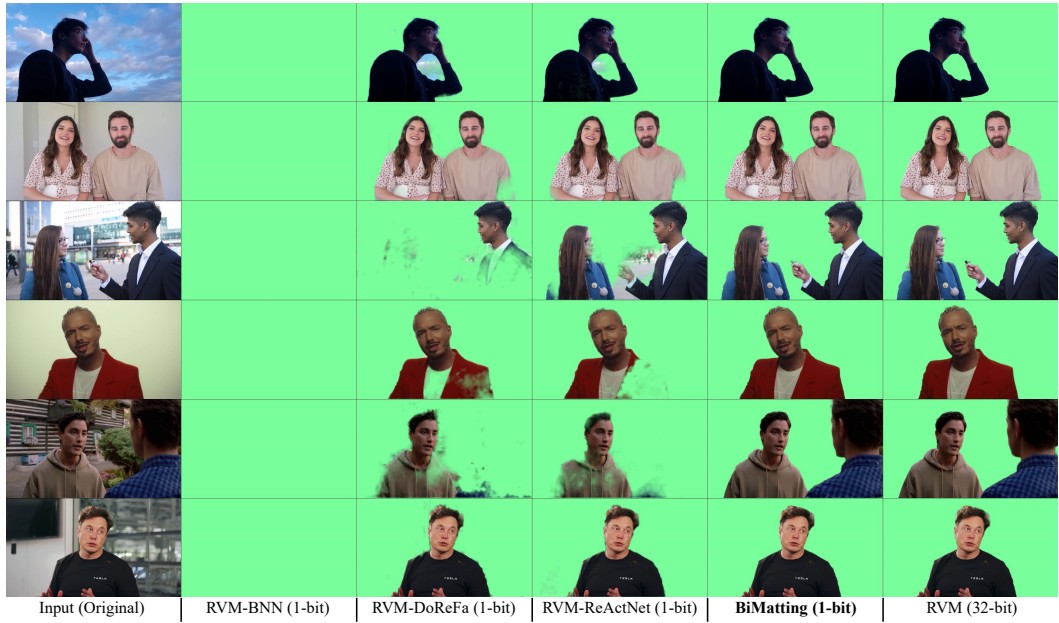

| Input (Original) | RVM-BNN (1-bit) | RVM-DoReFa (1-bit) | RVM-ReActNet (1-bit) | **BiMatting (1-bit)** | RVM (32-bit) |

Figure 6: More visual results. Compared to 1-bit video matting models using existing binarization methods, our BiMatting significantly surpasses them and achieves near full-precision performance. Note that the results of RVM-BNN indicate the model fully crashes.

For other parts, the Deep Guided Filter (DGF) incorporates a limited number of binarized $1 \times 1$ convolutions internally. For more detailed specifications, please refer to [40, 1]. The complete network is constructed and trained using PyTorch [50].

# 8 Additional Visualizations

## 8.1 Visual Results

We show more visual results in Fig. 6, where we can more clearly see the advantages of our BiMatting over other binarization methods, both in edge details and local region matting. At the same time, we also provide a video (BiMatting.mp4 file in the supplementary material) to show the advantages of our BiMatting in more detail.

## 8.2 Composited Datasets

We follow [1] as a guide to constructing composite training and test samples. We show some examples of composited training samples from the matting datasets in Fig. 7. The clips contain natural movements when compositing with videos as well as artificial movements generated by the motion augmentation. Motion augmentation was exclusively applied to the foreground and background of the image in the testing samples (Fig. 8). The motion augmentation solely involved affine transforms. Moreover, the strength of the augmentation was deliberately toned down in comparison to the training augmentation, ensuring that testing samples possess a high-degree realism.

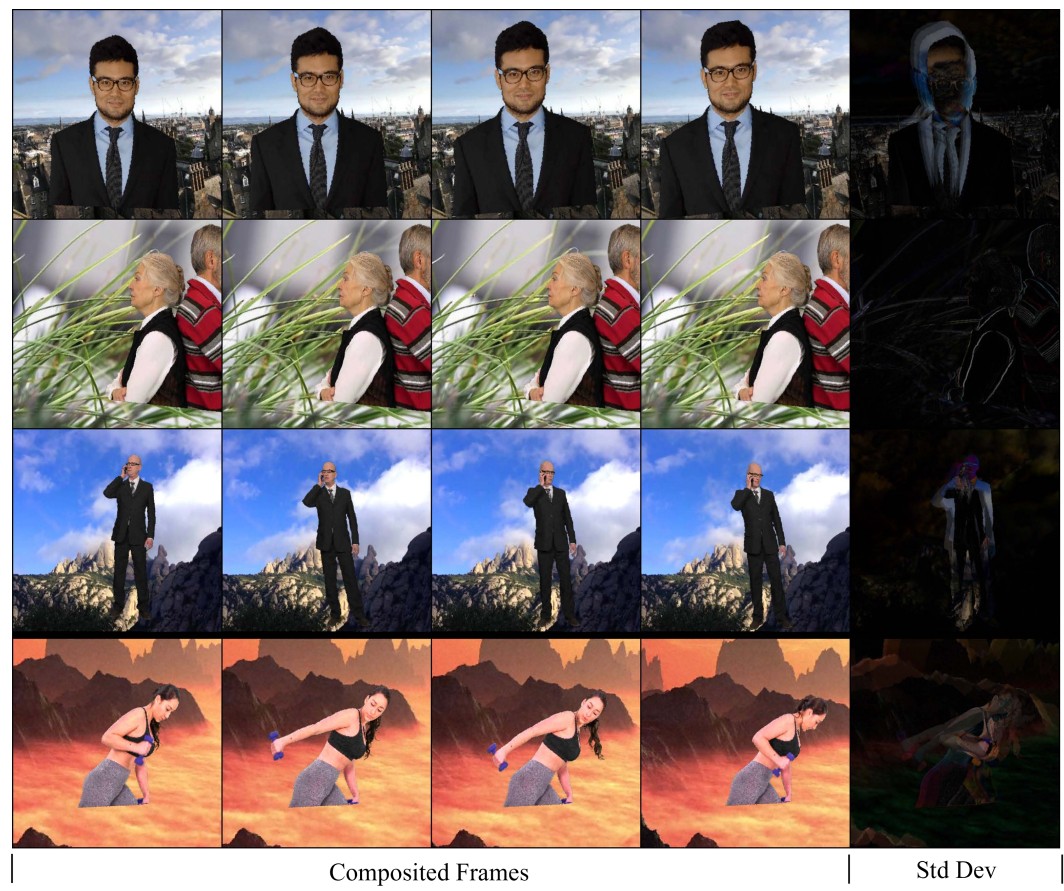

Composited Frames      Std Dev

Figure 7: Composite training samples. The last column is the pixels' temporal standard deviation.

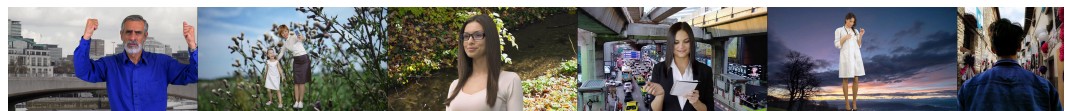

Figure 8: Example testing samples.

