# OpenReview forum: "BiMatting: Efficient Video Matting via Binarization"
_NeurIPS.cc/2023/Conference — NeurIPS 2023 poster_

### Official Review · Reviewer_N9ej · 2023-06-27

**Soundness:** 3 good
**Presentation:** 4 excellent
**Contribution:** 4 excellent
**Rating:** 8
**Confidence:** 5

**Summary:**

The authors propose the first binarized video matting network, namely BiMatting. They first analyze the bottlenecks of the direct binarization of video matting models and propose an accurate and efficient binarization method. Compared with other full-precision neural networks and other binarization methods, the authors confirm the effectiveness and great potential of BiMatting. It is worth noting this paper constructs a binary backbone that achieves similar acceleration compared to binarized mobilenetv3, a super-lightweight architecture, for the first time and realizes the practical application on video matting. The proposal and application of this binary network mean that the practicality of binarization has been significantly improved in general.

**Strengths:**

The authors propose the first binary neural network for video matting tasks for reducing computation consumption significantly while retaining practical accuracy. Since video matting tasks usually run on resource-constrained devices, this research is practically significant.

In terms of method, the authors successfully design an ultra-lightweight video matting network by binarization and make it have a significantly improved accuracy rate. In general, the authors' method tightly combines video matting tasks and architectures.

1. The binary backbone with SBB is interesting, and I think it is the major contribution of this paper. Compared with the existing binarization schemes, the authors create a backbone network with reasonable accuracy and comparable efficiency with the binarization mobilenetv3 through the careful design of the architecture. This is very important for the practical application of binary neural networks.

Moreover, the authors’ motivation (or paradigm) for designing the binary backbone network has a general contribution to the binarization community, that is, "the crucial paradigm of an accurate binarized encoder is the computation-dense form of binarized block", this motivation can explain the success of binary networks such as Bi-Real and IR-Net, and may lead to more lightweight binary networks with higher accuracy.

2. Using representations of different scales, SAB successfully uses sparse masks to further reduce the computational load of the binary decoder without significant performance degradation. This demonstrates that the representational ability of the binarized computational unit allows the model to exploit less informative representations.

The experimental and video results are solid. The accuracy of BiMatting not only far exceeds the binarized version of the existing model but also exceeds the full-precision matting model such as BGMv2 while achieving significant storage savings. The visualization results also show that BiMatting has improved significantly in detail.

**Weaknesses:**

1. For the backbone with SBB, the authors should provide more detailed information.
First of all, I suggest the authors give a more detailed ablation about the efficiency, including the FLOPs and the number of parameters of direct full-precision/binarized mobilenetv3 and SBB backbone.
In addition, since the authors add the pre-training process on the ImageNet dataset as stage 0, they should also provide the pre-training ablation results on this dataset, which has clarified the performance improvement of the binarized backbone alone.

2. For SAB, a strange phenomenon is that the results in Table 1 show that using SAB in the direct binarized matting model will cause a crash. Although SAB does not directly target the accuracy bottleneck, it even causes the model to be worse than direct binarization. Can the authors explain this phenomenon?

3. In Table 2, ReCU and BNN present the same collapse, however, as reported in their original papers, the former usually performs significantly better than the latter on ImageNet, even better than methods such as DoReFa-Net. I suggest the authors discuss this phenomenon and report the ImageNet pre-training accuracy of these binary beckbones.

4. I suggest the authors discuss the possibility of BiMatting's actual deployment, including how to deploy it on edge devices and possible hardware inference performance.

**Questions:**

Please see weaknesses

--After rebuttal--
Thanks for the detailed response, which has well addressed my concerns. I also read other reviewers' comments and the authors' responses. I am satisfied with the rebuttal and increase the rating score.

**Limitations:**

The authors discuss the limitations of their works in section 4.3.

---

> ### Author Rebuttal · Authors · 2023-08-09
>
> > **Q1**: For the backbone with SBB, the authors should provide more detailed information. First of all, I suggest the authors give a more detailed ablation about the efficiency, including the FLOPs and the number of parameters of direct full-precision/binarized mobilenetv3 and SBB backbone. In addition, since the authors add the pre-training process on the ImageNet dataset as stage 0, they should also provide the pre-training ablation results on this dataset, which has clarified the performance improvement of the binarized backbone alone.
>
> **A1**: We thank you for your attention and provide detailed information on BiMatting following your suggestions. We first compare the FLOPs and number of parameters of full-precision/binarized MobileNetV3 and SBB backbones, as well as their pre-training accuracy on ImageNet in Table A1. The results show that there will be severe performance degradation with the direct binarization of MobileNetV3, which directly causes the performance degradation of binarized video matting models. Our SBB backbone in BiMatting achieves significantly improved pre-training performance (56.1% vs. MBV3-BNN 24.62%) with 8.9x speedup and 21.4x compression ratio.
>
> Table A1 (*Table 4 of the attached PDF*): Pretrained results comparison on ImageNet.
> | Backbone              | \#Bit | \#FLOPs(G) | \#Param(M) | \#Accurary@1 |
> |--------------------|-------|-------------------|--------------------|--------------|
> | MBV3               | 32    | 1.07              | 11.34               | 63.00        |
> | MBV3-BNN           | 1     | 0.07              | 0.46               | 24.62        |
> | MBV3-DoReFa           | 1     | 0.07              | 0.46               | 22.38        |
> | MBV3-ReCU         | 1     | 0.08              | 0.53               | 32.76        |
> | SBB                | 1     | 0.12              | 0.53               | 56.09        |
>
> > **Q2**: For SAB, a strange phenomenon is that the results in Table 1 show that using SAB in the direct binarized matting model will cause a crash. Although SAB does not directly target the accuracy bottleneck, it even causes the model to be worse than direct binarization. Can the authors explain this phenomenon?
>
> **A2**: We hereby explain this phenomenon. As mentioned in our paper Sec 4.1 L279, the reason for the poor results of RVM + SAB is that SAB is designed to break the bottleneck of efficiency rather than accuracy, which leads to its cooperation with the original encoder or even results in poorer accuracy. Our SAB reduces the calculation of the decoder by masking the repeated intensive computation of continuous regions, which means that the remaining (not masked) regions in the feature need to provide enough effective information, and these features are provided by the encoder. Fig. 3 shows that when using the original encoder (directly binarized MobileNetV3), even if all other parts are restored to the full-precision counterparts, the performance still drops significantly. Therefore, the crashing can be expected when we use the original binarized encoder together with our efficient and lightweight (binarized) SAB decoder.
>
> > **Q3**: In Table 2, ReCU and BNN present the same collapse, however, as reported in their original papers, the former usually performs significantly better than the latter on ImageNet, even better than methods such as DoReFa-Net. I suggest the authors discuss this phenomenon and report the ImageNet pre-training accuracy of these binary backbones.
>
> **A3**: We further show the results of ImageNet pre-training in Table A1 according to the reviewer's suggestion. From these results, we can see that the backbone network of RVM-ReCU is significantly better than RVM-BNN. These results imply that the ReCU binarization method will fail in the video matting task, although it belongs to SOTA methods in the evaluation on ImageNet. Some existing work [1] also shows that the transfer of SOTA binarization methods on different tasks is not straightforward. For example, ReCU also leads to model collapse on 3D ShapeNet and GLUE benchmark tasks. It further verifies our motivation, *i.e.*, accurate and efficient binarization models should be tailored for the video matting task.
>
> [1] Qin H, et al. BiBench: Benchmarking and Analyzing Network Binarization. ICML 2023.
>
> > **Q4**: I suggest the authors discuss the possibility of BiMatting's actual deployment, including how to deploy it on edge devices and possible hardware inference performance.
>
> **A4**: The deployment of 1-bit BiMatting is supported by open-source libraries such as Larq [2] and daBNN [3] for ARM devices. Referring to the daBNN library and the performance of binarized operators implemented by it on the Raspberry Pi 4B, we believe that the binarized BiMatting can achieve similar speedup and compression on the hardware claimed in our paper.
>
> [2] Geiger L, Team P. Larq: An open-source library for training binarized neural networks. JOSS, 2020.
>
> [3] Zhang J, et al. dabnn: A super fast inference framework for binary neural networks on arm devices. ACM MM, 2019.

---

> > ### Comment · Reviewer_N9ej · 2023-08-15
> > **The response well adresses my concerns**
> >
> > I thank the authors for their detailed response, the authors put a lot of effort to answering the reviewers' questions. Considering all reviewers' comments and the author's responses, I can confirm that bimatting's contributions, especially its proposed lightweight binarization backbone, may have a broad impact on the field of binarization research in the future. The reproducibility of this work is also great, and I look forward to the authors releasing their complete training code and pre-trained models in their final version. So I would like to raise my score.

---

### Official Review · Reviewer_busB · 2023-07-06

**Soundness:** 3 good
**Presentation:** 2 fair
**Contribution:** 3 good
**Rating:** 6
**Confidence:** 4

**Summary:**

This paper propose an efficient solution that utilizes binarization to achieve real-time video matting on for devices constrained by computational resources. The proposed BiMatting constructs shrinkable and dense topologies of the binarized encoder block to enhance the extracted representation, while sparsifying the binarized units to reduce the low-information decoding computation. Extensive experiments shows that it outperforms SOTA binarized video matting methods by a large margin.

**Strengths:**

This work claimed itself as the first binarization solution for video matting tasks, which may provide a new effective solution to the real-time matting community.
The work is based on reasonable analysis and observation that shown in the Section 3.1. The proposed method also effectively address the pointed challenge. Moreover, although the performance is lower than the full-precision models, the proposed method achieves satisfactory results compared with the binarized video matting models.


**Weaknesses:**


1. The authors need to check the equations and make sure that all the notations are explained. For example, the ⊗ in Eq. (2) is not mentioned in the passage. The |W| is also remained unexplained.

2. Some key ideas are not fully verified by experiments. For examples, as the author claims that dense connection is very important to recover the performance, this assertion is somewhat unwarranted without the comparison of the proposed SBB and SBB w/o dense skip connections for each BiConv.

3. Since the Matting training phase has 4 stages, and each stage has its own independent training settings, the reviewer doubted that whether the Matting training phase can work well without extensive ablation study of different training settings. It would be better to provide experiments to demonstrate the effectiveness and robustness of the proposed method under different training settings. Otherwise, the robustness of the proposed method is hindered.


**Questions:**

See weaknesses.

---

> ### Author Rebuttal · Authors · 2023-08-09
>
> > **Q1**: The authors need to check the equations and make sure that all the notations are explained. For example, the ⊗ in Eq. (2) is not mentioned in the passage. The |W| is also remained unexplained.
>
> **A1**: $|{\mathbf{W}}|$ means to take the absolute value of the weight ${\mathbf{W}}$, $\otimes$ denotes the inner product of two binary vectors with bitwise XNOR and Bitcount operations. We will clarify our notation in the paper.
>
> > **Q2**: Some key ideas are not fully verified by experiments. For example, as the author claims that dense connection is very important to recover the performance, this assertion is somewhat unwarranted without the comparison of the proposed SBB and SBB w/o dense skip connections for each BiConv.
>
> **A2**: Thanks for pointing it out, we followed your suggestion to include more ablation experiments to show that dense connections of SBB are important to restore performance.
>
> We compared our BiMatting with the binarized model removing connections in SBB (BiMatting-NoConn) in *Table 2 of the attached PDF*. We find that after removing the connections, the performance of the binarized video matting model dropped significantly, which means that the connections in SBB are impossible. We also ablate the connection in SBB with the 1x1 binarized convolutions (BiMatting-BiConv) and summing up (BiMatting-SumConn), and then conduct the ablation experiments. The results in *Table 2 of the attached PDF* show it does not bring significant improvements while the binarized convolutions incur additional computation. The results further demonstrate the strengths of our proposed SBB, and we will update these results and discussion in our final version.
>
> > **Q3**: Since the Matting training phase has 4 stages, and each stage has its own independent training settings, the reviewer doubted that whether the Matting training phase can work well without extensive ablation study of different training settings. It would be better to provide experiments to demonstrate the effectiveness and robustness of the proposed method under different training settings. Otherwise, the robustness of the proposed method is hindered.
>
> **A3**: Please note that our training pipeline completely follows that of the baseline RVM, using the code from their public GitHub repository. We do not add additional training stages or other complications. We also adopt the same stopping conditions as RVM for a fair comparison. We also provide the detailed training pipeline of BiMatting in our General Response and will release our training code in the final version.
>
> We also provide the accuracy of checkpoints at the end of every training stage in Table A3. The results show that the accuracy of BiMatting is steadily increasing at each stage, and it already shows obvious advantages in the first few stages over existing binarized video matting models (RVM-BNN, RVM-DoReFa, RVM-ReCU, and RVM-ReAct). This phenomenon means that the results of our BiMatting are robust.
>
> Table A3 (*Table 3 of the attached PDF*): Low-resolution comparison on VM, D646, and AIM datasets for each stage.
> |         |           |       |       |       |       | Alpha |       |       | FG    |
> |---------|-----------|-------|-------|-------|-------|-------|-------|-------|-------|
> | Dataset | Method    | Stage | \#Bit | MAD   | MSE   | Grad  | Conn  | dtSSD | MSE   |
> | VM      | BiMatting (Ours) | 1     | 1     | 15.06 | 8.75  | 2.83  | 1.76  | 2.70  | -     |
> | 512x288 | BiMatting (Ours) | 2     | 1     | 13.50 | 7.02  | 3.32  | 1.52  | 2.69  | -     |
> |         | BiMatting (Ours) | 3     | 1     | 12.75 | 7.03  | 2.78  | 1.41  | 2.64  | -     |
> |         | BiMatting (Ours) | 4     | 1     | 12.82 | 6.65  | 2.97  | 1.42  | 2.69  | -     |
> | D646    | BiMatting (Ours) | 1     | 1     | 61.52 | 52.10 | 11.70 | 16.21 | 2.59  | 22.30 |
> | 512x512 | BiMatting (Ours) | 2     | 1     | 82.81 | 73.84 | 12.36 | 21.80 | 2.40  | 24.75 |
> |         | BiMatting (Ours) | 3     | 1     | 66.98 | 59.05 | 12.06 | 17.61 | 2.52  | 23.59 |
> |         | BiMatting (Ours) | 4     | 1     | 32.74 | 24.48 | 9.34  | 8.62  | 2.21  | 5.86  |
> | AIM     | BiMatting (Ours) | 1     | 1     | 54.26 | 44.24 | 13.31 | 14.30 | 2.41  | 23.40 |
> | 512x512 | BiMatting (Ours) | 2     | 1     | 61.19 | 51.44 | 13.99 | 16.11 | 2.21  | 24.12 |
> |         | BiMatting (Ours) | 3     | 1     | 63.19 | 53.88 | 13.69 | 16.59 | 2.30  | 20.23 |
> |         | BiMatting (Ours) | 4     | 1     | 35.17 | 26.53 | 9.42  | 9.24  | 1.82  | 7.00  |

---

> > ### Comment · Reviewer_busB · 2023-08-18
> >
> > I have read the responses and other reviewers' comments. Although some of my concerns are well addressed, the effectiveness of the proposed SBB and SAB is still weak in the original version. And the authors will add more comparison results and explanation. So I tend to accept this paper and keep my original rating score.

---

### Official Review · Reviewer_wDXB · 2023-07-06

**Soundness:** 3 good
**Presentation:** 4 excellent
**Contribution:** 3 good
**Rating:** 7
**Confidence:** 4

**Summary:**

The paper proposes a new video matting method called BiMatting. It is based on Binary neural networks (BNNs), a more compact network to reduce the computational and storage requirements of video matting. Specifically, the authors addressed the accuracy bottleneck of BNNs by re-designing its encoder and decoder architecture. Performance was evaluated on several video benchmarks and compared with SOTA methods.

**Strengths:**

This is the first time BNNs was used for video matting application. This problem was well-motivated and several challenges were addressed by careful architecture design and new training procedures. The authors provided a comprehensive comparison with other SOTA methods and demonstrated that this method offers a good tradeoff between performance and storage. It is computationally efficient that BiMatting reduces computational FLOPs by 11 times and storage by 21 times.

**Weaknesses:**

1. Although this method outperforms existing binarized video matting models, it is not yet on par with its full-precision counterpart in visual quality.
2. Another potential weakness is the complexity of the training pipeline of Bi-Matting, which contains a pre-training phrase and a matting training phrase with four stages.

**Questions:**

1. The authors claimed that binarizing the encoder causes a significant drop in performance. Why is that the case? Do the authors mean binarizing the activation of the encoder?
2. Is there a reason why the paper does not have a related work section? I would be curious to see recent works and other applications of BNNs to image segmentation and other relevant fields. It would help me evaluate the novelty and impact of current methods.

**Limitations:**

The author addressed the limitation that the method is not yet on par with 32-bit RVM. Quality-wise, it tends to get blurred and prefers simpler backgrounds.

---

> ### Author Rebuttal · Authors · 2023-08-09
>
> > **Q1**: Although this method outperforms existing binarized video matting models, it is not yet on par with its full-precision counterpart in visual quality.
>
> **A1**: As we present in our limitations paragraph and figures, BiMatting is not as accurate as full-precision models in certain highly dynamic scenes, due to the representation loss caused by the extreme bit-width compression. Nonetheless, our BiMatting significantly reduces the accuracy gap between the binarized video matting models and full-precision ones. Considering that BiMatting is the first binarization model in the video matting domain and has attained impressive gains in acceleration (12.4x) and compression efficiency (21.6x), we believe that the binarized video matting model holds substantial potential for further improved accuracy.
>
> > **Q2**: Another potential weakness is the complexity of the training pipeline of Bi-Matting, which contains a pre-training phrase and a matting training phrase with four stages.
>
> **A2**: Please note that our training pipeline completely follows that of the baseline RVM for a fair comparison, using the code from their public GitHub repository. We do not add additional training stages or other complications. We also adopt the same stopping conditions as RVM. We also provide the detailed training pipeline of BiMatting in our General Response and will release our training code in the final version.
>
> > **Q3**: The authors claimed that binarizing the encoder causes a significant drop in performance. Why is that the case? Do the authors mean binarizing the activation of the encoder?
>
> **A3**: We clarify here that the main reason for the significant performance drop when binarizing the encoder. It is caused by loss of representation capability induced by the coarse discretization (1-bit) of **both the model weights and activations**, especially when binarizing already lightweight architectures, like MobileNetV3. In the binarization process, 32-bit weights and activations are compressed to 1-bit. The representation capability and accuracy are therefore greatly reduced (from 2^32 to 2 possible states per weight or activation), which is the direct cause of the performance degradation of the binarized backbone. See Sec 2.1 and 3.1 for further discussion.
>
> > **Q4**: Is there a reason why the paper does not have a related work section? I would be curious to see recent works and other applications of BNNs to image segmentation and other relevant fields. It would help me evaluate the novelty and impact of current methods.
>
> **A4**: We discuss related work in Sec 2, including the related work on binarization (Sec 2.1) and video matting (Sec 2.2). We will follow the reviewer’s suggestion and add the section title of related work to make the manuscript more clear.
>
> We will also add a discussion on more related work on BNNs for image segmentation as suggested by the reviewer: Group-Net [1] demonstrates successful application to the semantic segmentation task on PASCAL VOC. Frickenstein et al. introduce Binary DAD-Net [2], the first BNN-based semantic segmentation network for drivable area detection in the autonomous driving field. Zhou et al. present CBNN [3], which incorporates multiple subnets with learnable global lateral paths and evaluates its performance on a segmentation dataset. However, the corresponding full-precision counterparts of these binarized networks are almost classical ResNet-18 architectures, and are thus not applicable to ultra light-weight architectures, such as MobileNetV3, which are susceptible to accuracy collapse when binarized. While image segmentation methods ultimately predict a discrete class, alpha matting requires the dense accurate prediction of a continuous alpha value. Segmentation methods are therefore not easily transferable to the video matting task.
>
> Please note that including those related works will not affect the main claims and findings of this paper.
>
> [1] Zhuang B, et al. Structured binary neural networks for accurate image classification and semantic segmentation. CVPR 2019.
>
> [2] Frickenstein A, et al. Binary dad-net: Binarized driveable area detection network for autonomous driving. ICRA, 2020.
>
> [3] Zhou X, et al. Cellular Binary Neural Network for Accurate Image Classification and Semantic Segmentation. IEEE TMM, 2022.

---

### Official Review · Reviewer_PTWg · 2023-07-07

**Soundness:** 3 good
**Presentation:** 3 good
**Contribution:** 3 good
**Rating:** 6
**Confidence:** 3

**Summary:**

The authors analyzed the operations inside the deep video matting networks and proposed an efficient binarization method to greatly reduce the computation cost. Specifically, they re-designed the encoder and also sparsified the decoding process. The proposed methods are shown to outperform the existing baselines and achieves reasonable visual quality.

**Strengths:**

- Section 3 shows binarization of the encoder parts bring the most harmful degradation of the accuracy, and the current decoder consumes the most computational resources. The analysis and preliminary experiments demonstrate the reason of the proposed methods. The logic and story-telling of the paper make a lot of sense.
- The review likes the reasoning of the encoder (lacking of short connection within the blocks) and decoder (many redundancy in computation), and the connections between the analysis and the proposed designs.
- The design of SAB makes sense for any video applications.

**Weaknesses:**

- The training pipeline is too complicated.
- Table 1 is confused. So for BiMatting (Ours), is it the same as RVM (SBB+SAB)? It's better to state the difference of RVM and BiMatting, and show it clearly in the table. Why RVM + SAB has much worse results than the one without any of SBB and SAB? Is it possible the bad performance is due to the bad training or need a better parameter set different from the proposed one?

**Questions:**

- In section 3.2.2, ep(5), the authors proposed to use mean to compute the values for 'short connection'. How did the authors come up with that solution? Why not using another convolution layers to reduce the channel number or taking the addition? Has the authors done some comparison?
- Any plans to extend the work to other video applications?

**Limitations:**

- The SAB designs may fail when the video is dynamic or having more foreground object movement.
- Complicated training procedures and vague stopping conditions for each stage. It makes the results in the paper hard to be reproduced.

---

> ### Author Rebuttal · Authors · 2023-08-03
>
> > **Q1**: The training pipeline is too complicated.
>
> **A1**: Please note that our training pipeline follows that of the baseline RVM for a fair comparison, using the code from their public GitHub repository. We do not add additional training stages or other complications. We also adopt the same stopping conditions as RVM. In addition, we provide the detailed training pipeline of BiMatting in our General Response and will release our training code in the final version. Please see the general response above for more details.
>
> > **Q2a**: Table 1 is confused. So for BiMatting (Ours), is it the same as RVM (SBB+SAB)? It's better to state the difference of RVM and BiMatting, and show it clearly in the table.
>
> **A2a**: Yes, BiMatting (ours) would be the same as "RVM (SBB+SAB)" in Table 1 of the original manuscript. To make the table more clear, we revise its notation, as shown in Table A2a. In the revised table, the checkmark and crossmark on the SBB column represent applying proposed SBB architecture and binarized MobileNetV3, respectively, and the checkmark and crossmark on the SAB column represent the applying SAB technique and direct binarization in the decoder, respectively. We will add this clarification in the next paper revision.
>
> Table A2a (*Table 1 of the attached PDF*): Ablation result of BiMatting on VM dataset.
> | SBB          | SAB          | \#Bit | \#FLOPs(G) | \#Param(MB) | MAD     | MSE    | Grad   | Conn   | dtSSD  |
> |--------------|--------------|-------|-------------------|--------------------|---------|--------|--------|--------|--------|
> | -            | -            | 32    | 4.57              | 14.5               | 6.08    | 1.47   | 0.88   | 0.41   | 1.36   |
> | ✕ | ✕ | 1     | 0.55              | 0.64             | 28.49   | 18.16  | 6.80   | 3.74   | 3.64   |
> | ✓   | ✕ | 1     | 0.57              | 0.67               | 14.81   | 7.63   | 3.16   | 1.70   | 2.70   |
> | ✕ | ✓   | 1     | 0.35            | 0.67               | 189.13  | 184.33 | 15.01  | 27.39  | 3.65   |
> | ✓   | ✓   | 1     | 0.37            | 0.67            | 12.82 | 6.65 | 2.97 | 1.44 | 2.69 |
>
> > **Q2b**: Why RVM + SAB has much worse results than the one without any of SBB and SAB?  Is it possible the bad performance is due to bad training or needs a better parameter set ...?
>
> **A2b**: Our SAB reduces the computation of the decoder by masking the repeated intensive computation of continuous regions, which means that the remaining (not masked) regions in the feature need to provide enough effective information, and these features are provided by the encoder. Fig. 3 shows that when using the original encoder (directly binarized MobileNetV3), even if all other parts are restored to the full-precision counterparts, the performance still drops significantly. Therefore, the crashing can be expected when we use the original binarized encoder together with our efficient and lightweight (binarized) SAB decoder. We will clarify this point by revising Sec. 4.1.
>
> In terms of settings, for fairness, we used exactly the same training settings and pipelines in all ablation experiments, which were completely consistent with the official code of the RVM baseline without any adjustments. This helps us reveal the real performance of our proposed techniques under equal settings.
>
> > **Q3**: In section 3.2.2, eq (5), the authors proposed to use mean to compute the values for 'short connection'. How did the authors come up with that solution? Why not using another convolution layers to reduce the channel number or taking the addition? Has the authors done some comparison?
>
> **A3**: As concluded from the analysis in Sec. 3.2.1, it is important to create a shortcut connection for each binarized convolution. We use the current form of connection for the following accuracy and efficiency considerations: (1) effectively recovering the representation and (2) constructing by parameter-free operations.
>
> For accuracy consideration, when we introduce an additional binarized convolution in connection to change the channel number, the introduced convolutions in connection also suffer representation degradation caused by binarization, which makes it hard to solve the degradation of the original binarized unit. The results in *Table 2 in the attached PDF* (BiMatting-BiConv) also show that this method does not bring significant improvement over the no-connection method (BiMatting-NoConn). We also compared the direct addition method (BiMatting-SumConn) mentioned by the reviewers. Although it avoids the information loss caused by binarization, directly summing up the features leads to the loss of fine-grained representation, making it still inferior to BiMatting in accuracy.
>
> For the efficiency consideration, even though it is binarized, the convolution introduced in the connection still brings additional computation and storage burden (see *Table 2 in the attached PDF*).
>
> > **Q4**: Any plans to extend the work to other video applications?
>
> **A4**: In this paper, we focus on the video matting task that is challenging for binarization, because the video matting model requires predicting continuous alpha instead of binary segmentation mask, and it also has a wider range of real-time applications on mobile devices. In future work, we plan to apply our proposed methods to speed up other dense tasks such as video segmentation and depth estimation.
>
> > **Q5**: Limitation: The SAB designs may fail when the video is dynamic or having more foreground object movement.
>
> **A5**: As we mentioned in our limitation paragraph, BiMatting does not perform as well as full-precision models in some highly dynamic scenes, due to the representation loss caused by the extreme bit-width compression. However, considering BiMatting is the first binarization model in the video matting field and achieves 12.4x acceleration and 21.6x compression gains, we believe the binarized video matting model has significant potential for improving accuracy in future work.

---

### Author Rebuttal · Authors · 2023-08-09

We deeply appreciate all reviewers for the positive reviews and constructive feedback. All reviewers agree that our BiMatting is highly efficient and contributes to both video matting and binarization fields significantly. Your expertise and insightful comments greatly help us to further improve our paper.

**Training details**:

Here, we first clarify the training details of our BiMatting, which was asked by three reviewers (Reviewer PTWg, wDXB, and busB).

We want to emphasize that our training pipeline strictly follows the baseline Robust Video Matting (RVM) for a fair comparison, using the code provided in their public GitHub repository. In particular, the training configurations and commands for BiMatting are identical to those outlined in the train.py#L8-L75 of the repository, without any modifications. Furthermore, we have made the complete network definition of BiMatting available in the BiMatting_code/model folder in the supplementary materials, making it easy for others to train and reproduce our results.

Additionally, since the RVM baseline uses MobileNetV3 pre-trained on the ImageNet dataset by default (from PyTorch official), our SBB backbone of the BiMatting network is equivalently pre-trained on ImageNet. Thus, we do not add any additional training stages or other complications. In the final version, we will release our complete training code and saved checkpoints, further facilitating the training and reproduction of BiMatting.

In the following responses to each reviewer, we provide detailed answers to all the questions raised.

---

### Comment · Area_Chair_sDMQ · 2023-08-18
**Discussion period**

Dear reviewers PTWg and wDXB,

Please acknowledge authors' response by posting a message in the respective thread. Does it change your score or address your concerns/comments?

AC

---

### Decision · Program_Chairs · 2023-09-21

**Decision:**

Accept (poster)

**Comment:**

The paper proposes an efficient video matting approach. The reviewers highlighted the analysis done in the paper, substantial improvements in latency, and storage, while maintaining strong performance. Throughout the discussion the reviewers kept being positive about the paper and its contribution. The authors did a good job in addressing the questions and concerns during the discussion stage. Hence the paper got positive ratings from all the reviewers and acceptance is recommended. Congrats!